

# Methodological approaches for estimating populations of the endangered dhole *Cuon alpinus*

Girish A. Punjabi[1,2], Linnea Worsøe Havmøller[1,3], Rasmus Worsøe Havmøller[3], Dusit Ngoprasert[4] and Arjun Srivathsa[1,5,6]

[1] Dhole Working Group, IUCN/SCC Canid Specialist Group, The Recanati Kaplan Centre, Tubney House, Tubney, United Kingdom
[2] Wildlife Conservation Trust, Mafatlal Centre, Nariman Point, Mumbai, India
[3] Research and Collections, Natural History Museum of Denmark, University of Copenhagen, Copenhagen, Denmark
[4] Conservation Ecology Program, King Mongkut's University of Technology Thonburi, Bangkok, Thailand
[5] Wildlife Conservation Society - India, Bangalore, India
[6] National Centre for Biological Sciences, TIFR, GKVK campus, Bangalore, India

Corresponding author
Linnea Worsøe Havmøller,
linnea.havmoeller@snm.ku.dk

## ABSTRACT

Large carnivores are important for maintaining ecosystem integrity and attract much research and conservation interest. For most carnivore species, estimating population density or abundance is challenging because they do not have unique markings for individual identification. This hinders status assessments for many threatened species, and calls for testing new methodological approaches. We examined past efforts to assess the population status of the endangered dhole (*Cuon alpinus*), and explored the application of a suite of recently developed models for estimating their populations using camera-trap data from India's Western Ghats. We compared the performance of Site-Based Abundance (SBA), Space-to-Event (STE), and Time-to-Event (TTE) models against current knowledge of their population size in the area. We also applied two of these models (TTE and STE) to the co-occurring leopard (*Panthera pardus*), for which density estimates were available from Spatially Explicit Capture–Recapture (SECR) models, so as to simultaneously validate the accuracy of estimates for one marked and one unmarked species. Our review of literature ($n = 38$) showed that most assessments of dhole populations involved crude indices (relative abundance index; RAI) or estimates of occupancy and area of suitable habitat; very few studies attempted to estimate populations. Based on empirical data from our field surveys, the TTE and SBA models overestimated dhole population size beyond ecologically plausible limits, but the STE model produced reliable estimates for both the species. Our findings suggest that it is difficult to estimate population sizes of unmarked species when model assumptions are not fully met and data are sparse, which are commonplace for most ecological surveys in the tropics. Based on our assessment, we propose that practitioners who have access to photo-encounter data on dholes across Asia test old and new analytical approaches to increase the overall knowledge-base on the species, and contribute towards conservation monitoring of this endangered carnivore.

## INTRODUCTION

Mammalian large carnivores are ecologically important, socio-culturally valued, and generally attract substantial conservation funds and resources for their protection (*Treves & Karanth, 2003*; *Dickman, Macdonald & Macdonald, 2011*). Reliable estimates of population size, while being crucial indicators of their status in the wild, remain difficult to obtain, particularly for species that are threatened with various degrees of extinction (*Kelly et al., 2012*; *Dröge et al., 2020*). For carnivores that typically occur at low densities and are distributed across large geographical regions, designing and executing scale-appropriate surveys to obtain such numbers also presents many challenges (*Royle, Stanley & Lukacs, 2008*; *Boitani, Ciucci & Mortelliti, 2012*). Inherently low populations of these species often yield sparse data, further constrained by logistical and administrative limitations (*Murphy et al., 2018a*; *Van der Weyde et al., 2021*). These issues can hamper and compromise the implementation of successful conservation monitoring and species recovery efforts.

Non-invasive sampling for population estimation has gained substantial traction among wildlife scientists and managers due to the ethical concerns with invasive capture-handling of carnivores, as well as the rapid development of less intrusive methods, including but not limited to indirect sign surveys coupled with occupancy-based models, genetic sampling, and camera-trap surveys (*Kelly et al., 2012*). While most of these methods are readily applicable to species with individually identifiable morphological traits (stripes, spots, rosettes, or other pelage patterns), recent methodological advances have allowed for exploring potential approaches to estimate populations of partially-marked or fully unmarked carnivore species (*Burgar et al., 2018*; *Forsyth, Ramsey & Woodford, 2019*; *Rich et al., 2019*). Nonetheless, some key challenges remain with genetic methods (*Mumma et al., 2015*; *Murphy et al., 2018b*) and camera-trap sampling (*Gilbert et al., 2021*) that practitioners need to be cognizant of before their application to species without natural markings. This is especially relevant for endangered carnivores, where under- or over-estimation of populations can result in erroneous conservation interventions (*Johansson et al., 2020*).

The Asiatic wild dog or dhole (*Cuon alpinus*) is among the top carnivores of Asia's tropical forest systems (*Kamler et al., 2015*). The social canid, currently on the IUCN Red List's Endangered category, is found across central Asia, the Indian subcontinent and Southeast Asia (*Kao et al., 2020*). Dholes were once distributed across large parts of Asia; their range is purported to have seen drastic contraction of nearly 80% in the last 100 years (*Wolf & Ripple, 2017*). Although the spatial extent of their current distribution is fairly expansive, most dhole populations are restricted to protected forest habitats (*Karanth et al., 2009*; *Jenks et al., 2012*; *Thinley et al., 2021*). While circumstantial evidence points to population declines in recent decades, literature on reliable population estimates to substantiate these assertions is virtually non-existent (*Srivathsa et al., 2020a*). This may be attributed to the fact that dholes do not have unique pelage patterns, precluding the ability to generate population estimates from camera-trap surveys—which has otherwise been commonplace for many species in the tropics—using standard mark–recapture methods (*Pollock et al., 1990*; *Nichols, 1992*; *Royle et al., 2014*). Dhole population status in the wild
has mostly been determined based on distribution assessments across various landscapes (*Srivathsa et al., 2014*; *Kamler et al., 2015*), except for some relatively recent studies that have sought to apply dedicated population models to estimate their population size and densities (*Selvan et al., 2014*; *Ngoprasert, Gale & Tyre, 2019*; *Srivathsa et al., 2021*).

Despite the ecological importance and conservation significance of dholes, attempts to estimate their population status in the wild remain woefully sparse. In this study, we had two broad objectives: (1) provide a synthesis of field and analytical methods that have been applied to determine the status of dhole populations in the wild—ranging from crude abundance indices to more sophisticated model-based estimators; and (2) apply a suite of recently developed estimation methods to dhole photo-encounter data obtained from camera-trap surveys in India's Western Ghats. Finally, we deliberate on the current state of knowledge regarding dhole populations across the species' distribution range, and provide recommendations for future research so as to optimize information gained from both, studies focused on dholes, as well as surveys where dhole data are generated as a by-product.

## MATERIAL AND METHODS

### Previous approaches to assess dhole population status

We used the terms "dhole + *Cuon alpinus* + population abundance + density + distribution + occupancy + relative abundance" on Google Scholar and searched for journal articles which either focused exclusively on dholes or those that included any approaches to assess dhole status as part of a larger mammalian assemblage from south Asia. Previous approaches to estimate dhole populations broadly fit into four categories—relative abundance indices (RAIs), distribution and occupancy, site-based abundance estimators (*i.e.,* population size estimated from occupancy-based models), or capture–recapture based estimators. For our synthesis, we only included those studies where one or more metrics reflecting dhole population status were available, either as RAIs, probability of occupancy or percentage distribution, abundance, or density estimates. Given the general paucity of dhole population studies, the first 30 search pages were adequate for undertaking a thorough review on the subject; although, some journal articles that were not indexed on the search engine or part of the initial search list were added at a later stage. We also included the IUCN Red List species evaluation (*Kamler et al., 2015*) to our synthesis as it contained dhole abundance estimates for one prominent landscape within the species' distribution range. None of the studies used more than one analytical approach. We summarised the results from each of these categories to map the geographical spread of past studies (Fig. 1), and understand the utility of these estimates for management and conservation of dholes.

### Empirical study in India
#### *Study area and design*

Radhanagari Wildlife Sanctuary covers an area of 351 km$^2$ and is located in the northern Western Ghats region of Maharashtra state, India (Fig. 2). The vegetation is dominated by tropical semi-evergreen and moist deciduous forests, interspersed by agriculture and two large reservoirs that supply water to the Kolhapur district. There are 20 villages inside

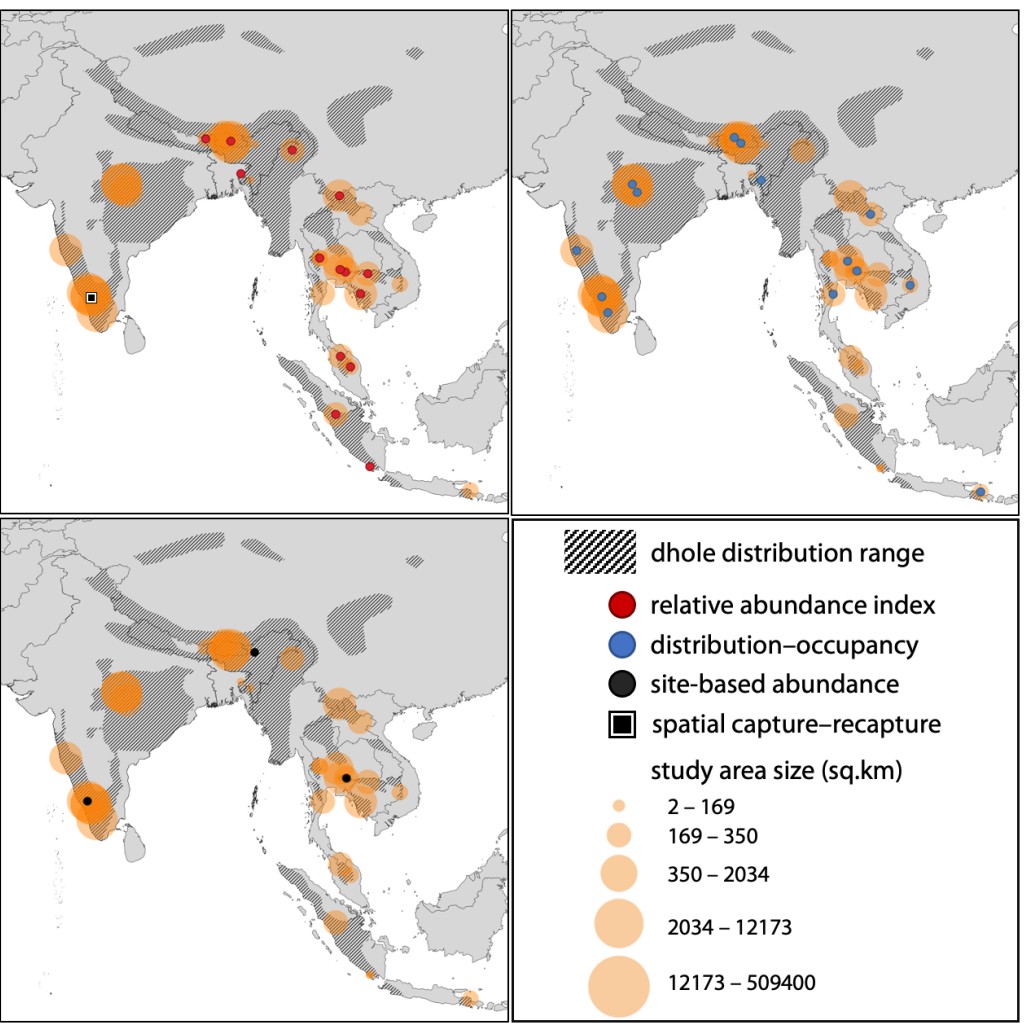

**Figure 1 Spatial locations of studies pertaining to population assessments of the dhole across the species' distribution range.** Spatial locations of studies pertaining to population assessments of the dhole across the species' distribution range ($n = 38$). The sizes of circles reflect the relative sizes of the study areas. All three panels include all study areas, but the point locations have been separated out based on the methodology used (RAI, occupancy/distribution, site-based abundance, or spatial capture–recapture) for ease of interpretation.

the sanctuary, and 20 more villages that border the sanctuary on all sides. The sanctuary is part of the larger Western Ghats landscape, a serially-listed UNESCO World Heritage site and an important Protected Area for the conservation of threatened fauna which include the leopard (*Panthera pardus*), dhole, sloth bear (*Melursus ursinus*), gaur (*Bos gaurus*), and sambar deer (*Rusa unicolor*).

We overlaid a grid-array of 80 cells of square geometry across the study area, with each cell measuring four km². Grid-cells with <20% forest cover ($n = 16$) were excluded from the study as they were considered unsuitable for supporting resident populations of the focal large mammals. This grid array was used to guide our surveys with the aim of placing

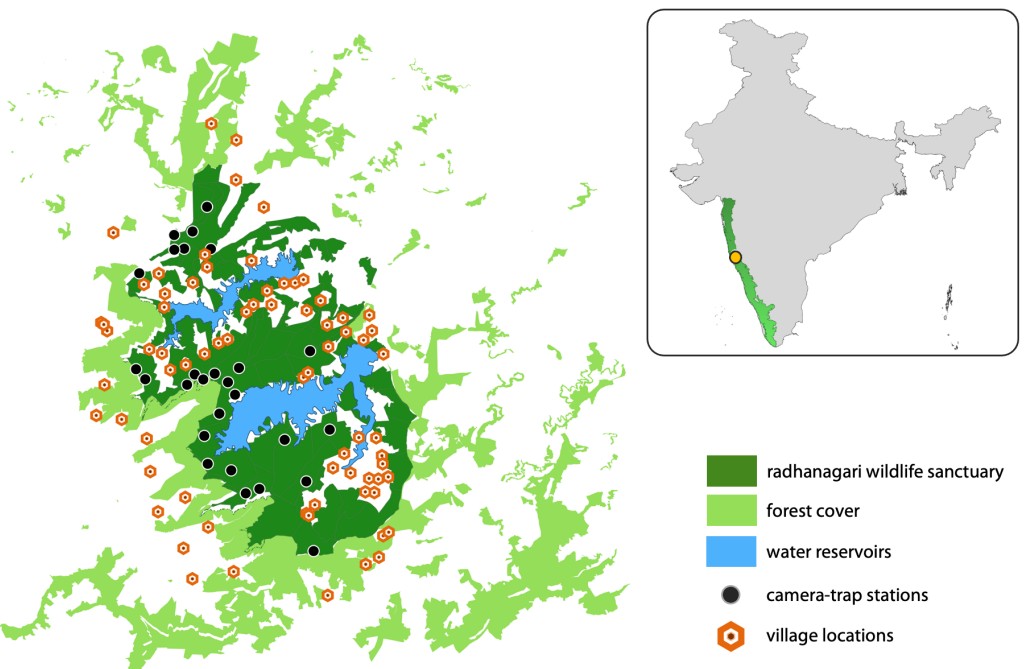

**Figure 2** **Study area map showing Radhanagari Wildlife Sanctuary.** Study area map showing Radhanagari Wildlife Sanctuary (351 km²), with the surrounding forest–non-forest habitat matrix, large water reservoirs, locations of camera traps and villages. Camera trap surveys were conducted in April–May 2019. Inset shows location of the study area within the larger Western Ghats landscape in India.

multiple camera stations within the home range of a dhole pack (∼25–202 km²; *Acharya, 2007*). Camera traps were set up at 34 locations; most cells had one camera-trap station (a pair of camera traps placed facing each other), and three cells had two camera-trap stations. Camera traps were placed based on information from sign survey data collected during a previous study in the landscape (*Punjabi et al., 2017*), knowledge of field staff, and our observations of animal signs during deployment. Practical difficulties did not allow for some areas to be sampled, but every effort was made to cover as much of the Protected Area as possible. At each station, a pair of camera-traps were placed at a height of 45–50 cm from the ground, facing each other at a distance of 1.5–2 m on either side of a forest road or trail. The average inter-station distance was 2 km (range = 0.8–5.8 km). Camera-traps were kept operational throughout the day (24-hr) for 16–18 days in 'blocks' (following *Royle et al., 2009*). Each camera trap was set to take one picture, followed by a 10-second video for every trigger event. The study was conducted in three 'blocks' (*i.e.*, not all locations had active cameras at the same time) from April to May 2019. Of the 34 locations, cameras from seven locations were lost due to theft, and the data could not be retrieved from these sites. The entire study was conducted in a short span of less than two months to reduce potential violation of the closure assumption. Camera coverage of the study region was limited (27 stations), as camera theft and time constraints were major impediments during the study.

The camera-trap data were curated on program Camera Base (Version 1.7) and capture histories of animal detections were extracted and compiled for each camera-trap station. Although assigning individual identities was not possible for dholes, some packs could be identified based on group size and number of adult/young members in the group when photo-captured on spatially adjacent camera-sites; these packs were assigned unique identities. The surveys also generated photo-captures of leopards. Since individual leopards could be uniquely identified based on the rosette marks on their pelage, they were assigned individual identities. The statistical models we intended to apply to dhole data are relatively new (described in detail below). We therefore estimated leopard population size and density using more established methods (Spatially Explicit Capture–Recapture 'SECR' models using Bayesian and Likelihood-based frameworks; see *Borchers & Efford, 2008*; *Royle et al., 2009*) as a 'control'–in some sense. We then fitted a set of models to both, the dhole encounter data and leopard encounter data, whilst comparing the leopard estimates from these models to the SECR-based estimates.

## Statistical methods to estimate dhole population size
### Direct counts
Wildlife managers sometimes rely on direct counts of species to make inferences on population size (*Gese, 2001*). But these counts can be problematic because they do not account for imperfect detection, double counting, and other sources of variation in the sampling/observation process, all of which are commonplace in ecological studies. Where species can be individually identified, practitioners sometimes use the minimum number of unique individuals as a crude measure of population size. The statistical unreliability of such counts notwithstanding, we calculated the minimum number of dholes for the sole purpose of making comparisons with other estimator-based numbers. We were able to identify three unique dhole packs with reasonable confidence; we calculated the minimum number of dholes in the region as counts of individuals from these three packs. Likewise, for leopards, we summed all the identified individuals. We use these minimum counts solely for the purpose of reasonably making an educated guess about the population size of the two species.

### Spatially-explicit capture recapture (SECR)
The use of SECR models has gained wide traction for estimating wildlife populations, given their relative advantages in estimation of density over conventional non-spatial capture–recapture models. Both, likelihood and Bayesian formulations of these models use spatial information in the capture data and also circumvent the issues with post-hoc calculation of sampled area for deriving density estimates (*Borchers & Efford, 2008*; *Efford, Borchers & Byrom, 2009*; *Royle et al., 2009*). Since the basic SECR model requires all individuals to be identified with certainty, we used this approach only for the leopard data to estimate abundance and density. For the SECR-Likelihood models, we used package *secr* in R (Version 4.1.1, R Development Core Team). For the SECR-Bayesian model, we used the package SPACECAP (*Gopalaswamy et al., 2012*), implemented in R. Additional model details and specifications are provided in Supplementary Information 1.

### Site-based abundance models

Site-based abundance models have piqued the interest of ecologists for estimating animal populations, particularly for species that do not have individually identifiable morphological traits (see *Gilbert et al., 2021*). The Royle-Nichols model (*Royle & Nichols, 2003*) was developed to derive abundance estimates from detection/non-detection data, while the N-mixture model relies on count data of animals for estimating abundance (*Royle, 2004*). Both models assume sampling sites are spatially independent, which is generally a difficult assumption to meet in practice. Advances in these methodological approaches include Bayesian implementations of beta-binomial mixture models which factor in spatial correlation of count data (*Martin et al., 2011*), and detection/non-detection data (*Chandler & Royle, 2013*; *Ramsey, Caley & Robley, 2015*).

Since count data were more informative, we used the *Martin et al. (2011)* approach for dholes (the leopard dataset did not meet the data–format requirements). The model assumes that the number of individuals at a site follows a Poisson distribution with $\lambda$ representing the average number of individuals per site. The subset of 'observed' or detected individuals $C$ is assumed to follow a binomial distribution of size $N$, with a detection probability $p$. Since individual detections are not independent, $p$ is modeled as a beta distribution. Mathematically, these may be expressed as:

$N_i \sim$ Poisson $(\lambda)$

$C_{ij} \sim$ binomial $(N_i, p_{ij})$

$p_{ij} \sim$ beta $(\alpha, \beta)$

Here, $i$ is the location, $j$ is the occasion, $\alpha$ and $\beta$ are the shape parameters of the beta distribution. Further, $\rho$ (correlation coefficient: magnitude of non-independence of detections, *i.e.*, autocorrelation) and p_ab (detection probability) are derived parameters calculated as $\rho = 1/(\alpha+\beta+1)$ and p_ab $= \alpha/(\alpha+\beta)$. We specified priors for $N$ to follow a gamma distribution, and tested for three scenarios–fully uninformed prior (gamma(0.01, 0.01)), partially informed prior (gamma (5, 3)) and a constrained prior (gamma (20, 15)). All models were run on JAGS implemented through R, with 10,000 iterations, five Markov chains, a burn-in of 2000, and the thinning rate set to one. We checked for model convergence through visual inspection of trace plots and the R-hat value (R-hat value closer to 1 indicates convergence; *Gelman & Hill, 2006*). The full R code is provided in Supplementary File 2.

### Time-to-Event (TTE) and Space-to-Event (STE) models

*Moeller, Lukacs & Horne (2018)* developed models for estimating abundances of unmarked animals from photo-encounter data using the camera-trapping rate and area of the camera viewshed. The TTE model is based on the timing of captures, but requires independent estimates of animal movement rate. The STE model is not sensitive to movement rates, as it substitutes space for time. Both models require the data to meet certain basic assumptions such as geographic and demographic closure, random placement of traps, and independence in animal observations. But we note that these models have been applied to studies where surveys were conducted with non-random placement of the camera trap, focusing on territorial species (*e.g.*, *Loonam et al., 2021*). These models

require measurements of area of the camera viewshed, which we calculated post-hoc using approximate measurements taken in the field. The area of a quadrilateral or a trapezium was calculated for both camera viewsheds at a station, within which the detection probability was assumed to be one (perfect detection). We note that this assumption may not always hold true as detection probability may vary due to different factors (*e.g.*, distance from the camera-trap). But the development of TTE and STE models being relatively recent, their current formulations do not allow for modelling imperfect detection at each site (*Moeller, Lukacs & Horne, 2018*). However, for a short distance from the camera-trap, if the animal enters the viewshed, the detection may be assumed to be perfect (*Moeller, Lukacs & Horne, 2018*). Since the area was calculated for both cameras at a station, we used the larger of the two viewshed areas (henceforth 'largest area') for the analysis. We also considered two additional scenarios whereby we increased this largest area by 10% and 20% to assess the effect of increased viewshed area on estimates from the TTE and STE models. We did so because the viewshed area could be potentially larger, as there were two camera units placed on opposite sides of a trail at each station.

For the TTE model, we used three hourly movement rates for leopards from three telemetry studies in India and Africa. For the dhole, we used three hourly movement rates from two studies, one in India and the other in Thailand (Supplementary File 2) to set the sampling period. The sampling occasions for all TTE models were considered to be one-hour periods, which contained sampling periods based on the hourly movement rate for each species. For each camera station, we sampled every two hours, beginning from the start to the end of the study period. For the STE model, we set the sampling frequency and sampling length as one second each, for both the species. In other words, any photo-encounter of a leopard or dhole during one second was considered as an independent detection. This was more computationally intensive, but ensured that sampling was instantaneous. Both the TTE and STE models were run under three different viewshed area scenarios as explained earlier—- largest area, largest area+10%, largest area+20%. All analyses were implemented in R (Version 4.1.1, R Development Core Team) through RStudio (Version 1.4.1717) using the package "spaceNtime" (*Moeller, Lukacs & Horne, 2018*). The R code is provided in Supplementary File 2.

## RESULTS

### Current knowledge of range wide dhole population status

Through our literature searches, we found a total of 38 studies that used one of the four broad categories (RAIs, distribution and occupancy, site-based abundance, or capture-recapture) to assess dhole population (Supplementary File 3). Of these, 17 studies (45%) used RAIs, while 17 studies (45%) used distribution and occupancy. Three studies (8%) used site-based abundance estimators, and only one study used capture–recapture based models (Fig. 2).

Study area sizes in reviewed literature ranged from 80 to 509,400 km$^2$. RAIs were generally calculated for 100 trap nights, except one study that used 1000 trap nights. RAIs for dhole varied from 0.008 to 7.41 across the 17 studies. Of the 17 other studies that dealt

with distribution models, 14 used probability of occupancy or percentage occurrence as a metric, whereas three studies used MaxEnt models to calculate habitat suitability (area) for dholes. Estimates of reported occupancy probabilities (means) varied from 0.12 to 0.95, while suitable habitat area varied from 7% to 72% depending on the study area size. Of the three studies that used site-based abundance approaches, one estimated 207–304 individuals across a 37,000 km$^2$ landscape of the Western Ghats, while another study estimated site-level abundance at 0.26 ($\pm$0.02) equating to a density of 6.62 ($\pm$0.58) dholes per 100 km$^2$ in a Protected Area of India's Eastern Himalayas. The third study estimated the density to range from 2.2 to 3 dholes per 100 km$^2$ in two protected areas of Thailand. The lone study that applied spatial capture–recapture models used a combination of genotype-based individual identification and indirect signs, and the estimated density ranged from 12 to 14.2 dholes per 100 km$^2$ in one Protected Area of India's Western Ghats (Fig. 2).

### Empirical study in India

Based on direct counts of uniquely identifiable individuals, we determined the minimum number of the focal animals in our study location was 29 dholes and nine leopards. From the SECR models (likelihood and Bayesian formulations), we estimated leopard population size at 11.84 to 12.53 individuals (see Fig. 3 and Supplementary File 1). As shown in Fig. 4, the site-based abundance models overestimated dhole population size by a large margin (considering the minimum number from direct counts and ecological expectation) across all scenarios (abundance range: 104–189 individuals; Supplementary File 2). Estimated detection probability was low across all models (<0.10) and the correlation parameter was positive but closer to zero. We found the STE models to be the most reliable in estimating population size and densities of dholes ($N = 33.03 \pm 4.07$; $D = 9.15/100$ km$^2$) and leopards ($14.31 \pm 2.68$; $D = 3.96/100$ km$^2$; Fig. 3). In particular, the STE model specified with the 'largest area' setting (see Methods section for details) produced the most precise estimates–relatively speaking–that were comparable to direct counts for both species, our field-knowledge based expectation from the area, and the SECR-based estimates for leopard. The TTE model did not perform well under any of the scenarios of varying movement rates (for either species), and the estimates of population size were unreasonably high and imprecise (Table 1).

## DISCUSSION

Our review of literature on dhole population status revealed that studies of robust population estimation across the species' range remain extremely scarce. This may be because dholes were not the primary species of interest in some studies, but also because they do not have distinct natural marks, thereby making it difficult to assign individual identities (such as with camera-trap photo-encounter data). It is likely that a majority of studies have therefore used RAIs, or assessed the distribution of dholes in terms of habitat occupancy or extent of suitable habitats. Only a small percentage of studies examined dhole population size using site-based abundance models (*Selvan et al., 2014*; *Kamler et al., 2015*; *Ngoprasert, Gale & Tyre, 2019*), or spatial capture–recapture using genetic methods

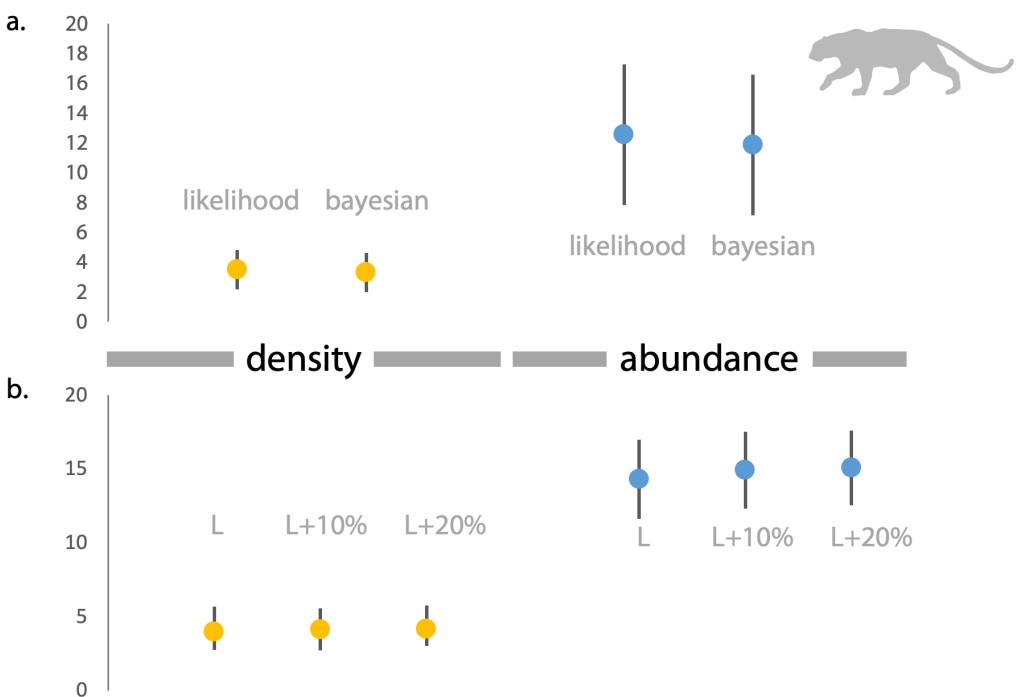

**Figure 3 Estimates of density and abundance of leopards in Radhanagari Wildlife Sanctuary.** (A) Estimates of density (left) and abundance (right) of leopards in Radhanagari Wildlife Sanctuary (April–May 2019), based on Spatially Explicit Capture–Recapture models. Density calculated as individuals per 100 km$^2$. Error bars for the likelihood-based estimates indicate Standard Errors, and those for Bayesian estimates are Standard Deviation values. (B) Estimates of density (left) and abundance (right) of leopards in Radhanagari Wildlife Sanctuary (April–May 2019), based on space-to-event models (Moeller et al. 2011). Density calculated as individuals per 100 km$^2$. L, L+10% and L+20% are three scenarios, where viewshed areas were calculated as 'largest', 'largest+10%' and 'largest+20%'. Error bars for density are 95% confidence intervals.

(*Srivathsa et al., 2021*). Our empirical study confirms that estimating dhole population size with camera-trap photographs is challenging, especially when model assumptions are not fully met or the data are scarce–both of which exemplify common issues with photo-encounter data of large carnivores in most tropical regions. We tested three models which did not require individual identification; one site-based abundance model that required count data (*Martin et al., 2011*), and two models that used species-specific camera-trapping rate (TTE and STE, *Moeller, Lukacs & Horne, 2018*). Of these, only the STE model, which was not sensitive to animal movement rate, showed some promise in generating reasonable population size estimates for dholes. STE model estimates for the leopard served as a useful control for the dhole as we had comparable estimates from SECR and direct counts.

Population size or density estimates of dholes are rare in the literature, even though they are valuable for conservation management at the scale of protected areas. Of the three models we tested to derive population estimates of dholes, the STE and TTE models are designed to work with random placement of camera-traps and time-lapse pictures to avoid variable detectability associated with motion-sensor pictures (*Moeller, Lukacs &*

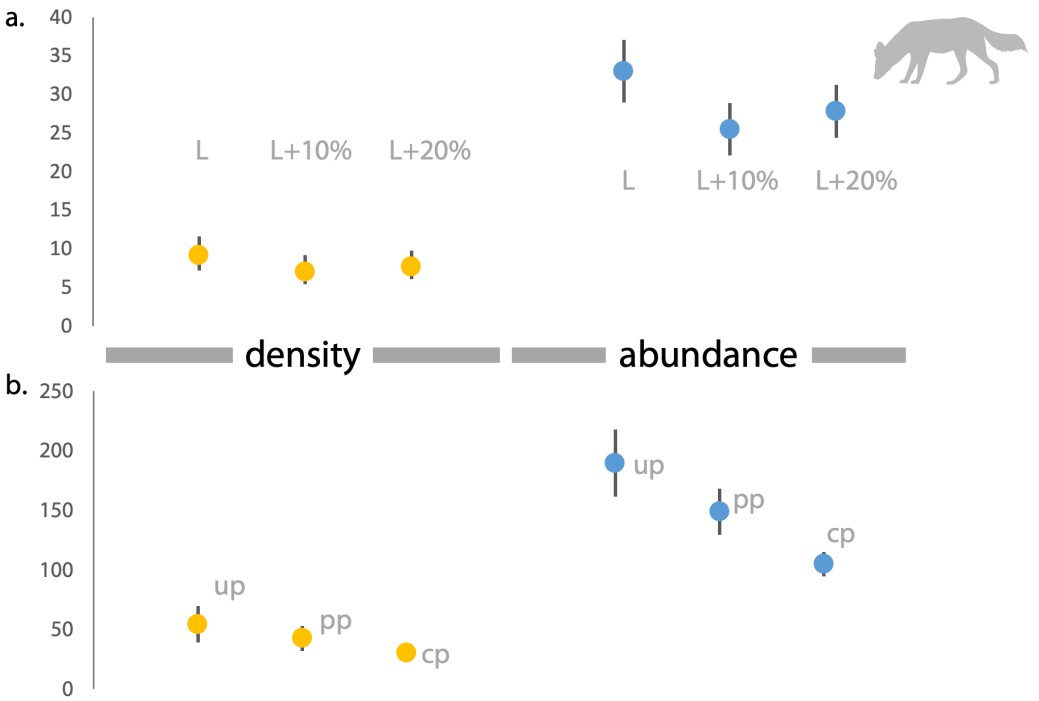

**Figure 4 Estimates of density and abundance of dholes based on space-to-event 'STE' and Site-based Abundance (SBA) models in Radhanagari Wildlife Sanctuary.** (A) Estimates of density (left) and abundance (right) of dholes in Radhanagari Wildlife Sanctuary (April–May 2019), based on space-to-event 'STE' models. (B) Estimates of density (left) and abundance (right) of dholes in Radhanagari Wildlife Sanctuary (April–May 2019), based on site-based abundance models (beta-binomal/Poisson mixture models; *Martin et al., 2011*). Density was calculated as individuals per 100 km². up–uninformed priors, pp–partially informed priors, cp–constrained priors. Error bars for density are 95%.

*Horne, 2018*). However, many carnivores are inherently rare and therefore camera-trap data on carnivores is generally collected from non-random placement of cameras along trails or roads to maximise encounter probability (*Karanth et al., 2017*; *Iannarilli et al., 2021*). Despite using a non-random survey design and motion-sensor pictures, we derived biologically reasonable estimates for both the dhole and leopard using the STE model. For the leopard, estimates from STE models matched those generated from SECR models. It is important to note here that settings for sampling frequency and length for the STE model were kept to 1 s each, representing a snapshot in time, as larger settings (not presented) resulted in overestimation (noted in *Loonam et al., 2021*). The TTE model, on the other hand, was not a useful approach for either species, as all the estimates were biased high across all scenarios of animal movement rates that we specified. We believe this may be due to two plausible reasons; first, as the TTE model shows high sensitivity to hourly movement rate for deriving sampling periods, the values we supplied were not representative of movement rates in our study region; and second, our study design used non-random camera placement and motion-sensor pictures that likely inflated trapping rate, reduced the number of sampling periods before first detection, and thereby overestimated the population size. Additionally, both the TTE and STE models rely on first detection of

**Table 1 Estimates of population abundance and density from the Time-to-Event models.** Estimates of population abundance and density from the Time-to-Event models (TTE) under different scenarios of hourly movement rates for the dhole and leopard in Radhanagari Wildlife Sanctuary from the Western Ghats, India.

| | Dhole | Leopard |
|---|---|---|
| **Model** | **Average hourly movement rate = 266 m/hr** | **Average hourly movement rate = 77.58 m/hr** |
| TTE (Largest area) | | |
| a. N ±SE (CI) | 372.67 ± 189.51 (145.57–954.05) | 1226.61 ± 409.14 (648.9–2318.33) |
| b. D (CI)/100 km² | 103.23 (40.32–264.28) | 339.78 (179.75–642.2) |
| TTE (Largest area + 10%) | | |
| a. N ±SE (CI) | 173.11 ± 123.89 (49.08–610.55) | 1132.22 ± 377.84 (598.87–2140.57) |
| b. D (CI)/100 km² | 47.95 (13.59–169.13) | 313.63 (165.89–592.96) |
| TTE (Largest area + 20%) | | |
| a. N ±SE (CI) | 329.83 ± 164.23 (131.12–829.66) | 703.07 ± 287.67 (325.16–1520.2) |
| b. D (CI)/100 km² | 91.37 (36.32–229.83) | 194.76 (90.07–421.11) |
| **Model** | **Average hourly movement rate = 424 m/hr** | **Average hourly movement rate = 200 m/hr** |
| TTE (Largest area) | | |
| a. N ±SE (CI) | 90.34 ± 90.43 (17.65–462.46) | 506.83 ± 227.05 (219.2–1171.9) |
| b. D (CI)/100 km² | 25.02 (4.89–128.11) | 140.4 (60.72–324.63) |
| TTE (Largest area + 10%) | | |
| a. N ±SE (CI) | 332.27 ± 165.80 (131.86–837.28) | 375.89 ± 187.67 (149.10–947.62) |
| b. D (CI)/100 km² | 92.04 (36.53–231.93) | 104.12 (41.30–262.5) |
| TTE (Largest area + 20%) | | |
| a. N ±SE (CI) | 77.15 ± 76.72 (15.19–391.86) | 521.21 ± 212.92 (241.33–1125.7) |
| b. D (CI)/100 km² | 21.37 (4.21–108.55) | 144.38 (66.85–311.83) |
| **Model** | **Average hourly movement rate = 791.7 m/hr** | **Average hourly movement rate = 264.37 m/hr** |
| TTE (Largest area) | | |
| a. N ±SE (CI) | 83.53 ± 84.65 (16.08–433.87) | 579.87 ± 237.11 (268.30–1253.24) |
| b. D (CI)/100 km² | 23.14 (4.54–120.19) | 160.63 (74.32–347.16) |
| TTE (Largest area + 10%) | | |
| a. N ±SE (CI) | 154.49 ± 109.78 (44.12–540.92) | 351.52 ± 176.51 (138.77–890.41) |
| b. D (CI)/100 km² | 42.79 (12.22–149.84) | 97.37 (38.44–246.65) |
| TTE (Largest area + 20%) | | |
| a. N ±SE (CI) | 354.73 ± 159.70 (152.83–823.34) | 329.89 ± 164.22 (131.18–829.62) |
| b. D (CI)/100 km² | 98.26 (42.34–228.07) | 91.38 (36.34–229.81) |

**Notes.**

Largest area, largest viewshed area; Largest area + 10%, largest viewshed area + 10% of viewshed area; Largest area + 20%, largest viewshed area + 20% of viewshed area; N, estimated abundance; SE, estimated standard error; CI, 95% confidence intervals; D/100 km², estimated density per 100 km².

the animal during a sampling occasion. Variable detection probability at the scale of the camera viewshed can also affect abundance estimates. Increasing area of the viewshed also negatively affects the estimates of abundance, as seen from our comparison of three different viewshed area (Largest area, +10%, +20%) scenarios. Therefore, viewshed area should be measured accurately in field settings to avoid biasing the abundance estimates.

The site-based abundance model (described by *Martin et al., 2011*) appeared to overestimate dhole population size even with informed priors. This is likely because the estimated detection probability of dholes was extremely low (<0.1 across all models)

in our study, which can positively bias the abundance estimate (*Williams, Nichols & Conroy, 2002*). The estimated correlation parameter ($\rho$) was positive, ranging from 0.10 to 0.11, suggesting non-independence of detections and some heterogeneity in detection probability (*Martin et al., 2011*). We recognize that our study design could have led to double counting of dholes across camera-trap sites, as sites may not have been sufficiently independent (inter-trap distance: 0.8–5.8 km). In their application of this model with certain modifications, *Ngoprasert, Gale & Tyre (2019)* thinned their data by removing trap locations that were within 2 km of each other to deal with this problem. In our case, the dataset was too sparse to begin with, making such post-hoc manipulations infeasible. To fully gauge its reliability, this model warrants further testing on larger camera-trap datasets where count data on dholes are correlated across sampling units. We also acknowledge that our limited sample size and the relatively homogeneous habitat (protected forest area) made it difficult to incorporate spatial covariates to examine site-level variations in local abundance and factors affecting density, as it has been done for *e.g.*, leopards (*Havmøller et al., 2019*). Future studies could explore how abundance of preferred prey species, presence of competing co-predators, and anthropogenic factors affect dhole spatial densities.

Taken together, our comparison of the three models reveals that the STE model can be a potential approach for camera-trap studies where dholes were the primary focus or bycatch data is collected on the species. The STE model should therefore be tested widely to assess its utility in deriving meaningful estimates for dholes, especially in scenarios when some model assumptions are not met. It would also be worthwhile to examine study designs with random sampling (off-trails) and time-lapse pictures to understand if and how these would affect trapping rate and corresponding population estimates. Study designs that use random placement of cameras and time-lapse pictures for rare carnivores will require a high number of camera traps and longer survey periods to accrue adequate sample sizes, which would entail higher operational costs and possible violations of the closure assumption. However, doing so may serve as a suitable alternative to index-based methods (RAIs) which are still fairly common in studies of dholes in South Asia, despite their widely acknowledged flaws (*Sollmann et al., 2013*).

Our results also have conservation relevance in light of recent findings from studies of dholes in South Asia. *Srivathsa et al. (2020a)* and *Srivathsa et al. (2020b)* found that dholes occupy nearly 249,606 km$^2$ of forest and agro-forest habitats, including 162 PAs, across India; yet population estimates from the majority of these PAs are unavailable. We found a density of ~9 dholes/100 km$^2$ in our study area, Radhanagari Wildlife Sanctuary, from the northern Western Ghats, which is lower than 12–14 dholes/100 km$^2$ estimated from Wayanad Wildlife Sanctuary in southern Western Ghats (*Srivathsa et al., 2021*). However, our estimate is higher than those reported from northeast India's Pakke Tiger Reserve (6–7 dholes/100 km$^2$; *Selvan et al., 2014*) and Thailand's Dong Phayayen-Khao Yai–Kaeng Krachan Forest Complex (2–3 dholes/100 km$^2$; *Ngoprasert, Gale & Tyre, 2019*). Radhanagari has moderate densities of herbivore ungulates and protection levels, and shares tenuous connectivity to other dhole habitats in the landscape (*Punjabi et al., 2017*; *Rodrigues, Srivathsa & Vasudev, 2021*). Three breeding packs were identified in our study area (pack size ranging from six to 12), which produces a conservative number of six

breeding adults in an estimated population of 33 dholes. Clearly, this population is very small and at a high risk of local extinction in a short span of 50 years if isolated due to break in connectivity (*Kao et al., 2020*). Considered together, the number of dhole breeding pairs, density, and abundance estimates in Radhanagari emphasizes the PA's high conservation value for the species, both in the Western Ghats landscape, and globally.

Current understanding of dhole ecology, social structure and adaptability is plagued by wide knowledge gaps in several aspects. In African wild dogs (*Lycaon pictus*)–a species that is phylogenetically and behaviorally similar to dholes–a minimum number of individuals is needed in a pack in order to successfully hunt and sustain breeding (*Courchamp & Macdonald, 2001*). Whether the same is true for dholes remains unknown. It is reasonable to assume that generating estimates of not just individuals, but the number of breeding packs per area, the average pack-size per area or pack densities, may be more ecologically relevant for dholes, as has been done with wolves (*Canis lupus*; *Mattioli et al., 2018*). Additionally, the continued decline of dholes and other large carnivores in Asia also calls for building upon population size estimates from select areas and progress towards understanding drivers of population dynamics and local extinctions across larger spatial and temporal scales. A recently developed transnational genetic registry has allowed for forecasting population dynamics for species like the wolf, brown bear (*Ursus arctos*) and wolverine (*Gulo gulo*) in Scandinavia (*Bischof et al., 2020*). A similar repository and approach for threatened large carnivores in Asia would serve as a crucial conservation tool and allow for planning long-term management strategies.

## CONCLUSION

Ecological data on carnivore populations are inherently noisy, and surveys often generate sparse detections of species which are either uncommon or occur at low densities (*Gese, 2001*; *Royle, Stanley & Lukacs, 2008*; *Boitani, Ciucci & Mortelliti, 2012*). When surveying unmarked animal populations, issues such as non-identifiability of individuals, spatial autocorrelation among sampling units, heterogeneity in animal detections, and sparse photo-captures affect the analytical approach and inferences drawn (see *Gilbert et al., 2021* for a review). Estimating populations of unmarked animals therefore presents a suite of methodological challenges. We find that the space-to-event models showed some promise in estimating dhole populations in our study region, likely because these models are not sensitive to the animal's movement rate. Other novel methods such as the REM (*Lucas et al., 2015*), REST model (*Nakashima, Fukasawa & Samejima, 2018*) or camera-trap based distance sampling (*Howe et al., 2017*) could not be explored in our study due to the inherently different study design and assumption requirements, but have been successfully demonstrated in other species under certain scenarios (*Palencia et al., 2021*). We encourage further development of user-friendly tools for implementing such analyses of data on unmarked species, as this is likely the reason for low publication rates in terms of population studies of dholes and similar unmarked species. The widespread application of camera-trapping methodology to study felids across Asia (*e.g.*, *Macdonald et al., 2019*) have led to large amounts of photo-encounter data being generated; these datasets likely include

photo-encounter data on dholes, but remain unutilized. We urge researchers with access to these data to (1) explore the applications detailed in this paper, and/or (2) make their data available on public repositories so as to permit carnivore researchers and statistical ecologists to evaluate old and newly developed models to estimate dhole populations, and thereby contribute towards conservation monitoring of this endangered species.

## ACKNOWLEDGEMENTS

We thank Mr. MK Rao, Mr. SM Gujar, late Mr. VR Khedkar and field officers and staff of the Maharashtra Forest Department in Radhanagari. We are grateful to N Songsasen and the IUCN Dhole Working Group for facilitating the initial meetings that eventually led to the conceptualization of this manuscript.

### Funding

The field study in Radhanagari Wildlife Sanctuary was funded and permitted by the Maharashtra Forest Department, Govt. of India. Girish A Punjabi was supported by a grant from Vinati Organics Limited (VOL) to Wildlife Conservation Trust during the writing of this manuscript. Linnea Worsøe Havmøller received funding from the European Union's Horizon 2020 research and innovation programme under the Marie Skłodowska-Curie grant agreement No. 801199. Rasmus Worsøe Havmøller was supported by research grant 36069 from VILLUM FONDEN. Arjun Srivathsa was supported by the Wildlife Conservation Network. There was no additional external funding received for this study. The funders had no role in study design, data collection and analysis, decision to publish, or preparation of the manuscript.

### Grant Disclosures

The following grant information was disclosed by the authors:
The Maharashtra Forest Department, Govt. of India.
Vinati Organics Limited (VOL) to Wildlife Conservation Trust.
The European Union's Horizon 2020 research and innovation programme.
VILLUM FONDEN.
The Wildlife Conservation Network.

### Competing Interests

The authors declare there are no competing interests.

### Author Contributions

- Girish A. Punjabi conceived and designed the experiments, performed the experiments, analyzed the data, prepared figures and/or tables, authored or reviewed drafts of the paper, and approved the final draft.
- Linnea Worsøe Havmøller and Arjun Srivathsa conceived and designed the experiments, performed the experiments, analyzed the data, prepared figures and/or tables, authored or reviewed drafts of the paper, and approved the final draft.

- Rasmus Worsøe Havmøller conceived and designed the experiments, performed the experiments, authored or reviewed drafts of the paper, and approved the final draft.
- Dusit Ngoprasert analyzed the data, authored or reviewed drafts of the paper, and approved the final draft.

## Data Availability

The R code for running the SECR analysis and the R code for running the site-based abundance model, Space-to-Event model and Time-to-Event model are available in the Supplementary Files.

## Supplemental Information

Supplemental information for this article can be found online at http://dx.doi.org/10.7717/peerj.12905#supplemental-information.

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
