# Peer review of "Methodological approaches for estimating populations of the endangered dhole Cuon alpinus"

_PeerJ, doi:10.7717/peerj.12905_

## Round 0.1 · original submission · Minor Revisions

This is a sound, well written and interesting paper that was appreciated by both referees. Both reviewers provide a number of useful suggestions for clarifying the methodology and context. The revisions required to address these critiques should not be arduous, and will improve the readability and general usefulness of the paper to others hoping to improve estimates of abundance of wild populations, and to encourage wider use of more statistically appropriate techniques that better account for common biases introduced by simple indices.

·

Basic reporting

The paper is clearly written, providing sufficiently detailed background and explanations. It is appropriately organised. The supplementary materials are sufficiently detailed to reproduce the analyses. (See Additional comments for specific recommendations.)

Experimental design

The research questions and rationale for the work are well defined/explained. Investigations are performed to a high technical and ethical standard. (See Additional comments for specific recommendations.)

Validity of the findings

Above and beyond the utility of the paper with respect to the surveying and the subsequent management of Dhole populations, the comparison of existing density models applied to new (and bench-marked) data sets is a useful contribution to the literature. The models appear to be specified and estimated correctly, using robust methodology. (See Additional comments for specific recommendations.)

Additional comments

General comments

This well-written paper, which includes both a review of relevant literature and an empirical study, addresses the important practical problem of estimating population densities for unmarked species with small population sizes. In the case of dholes (the species of interest), the authors provide a strong rationale for their work based the paucity of existing knowledge and the need to better inform conservation efforts.

Although many statistical methods have been developed for unmarked density estimation, it is not always obvious which method is most appropriate for a given study. The problem of selecting among candidate models is not just about goodness-of-fit or predictive scores, the validity of inferred quantities (e.g., density) depends on both the degree of violation of the underpinning assumptions of the model and the sensitivity of the model to these violations; yet, these dependencies are difficult to assess. To this end, the authors include a study of a second (marked) species as a bench mark for (unmarked) model performance, leading to useful model recommendations for practitioners working on similar studies.

Overall, I would recommend this paper for publication in PeerJ, given appropriate adjustments/responses to the specific comments/questions that I have added below.

Specific comments

1) Line 135: How many cells have >20% cover?
2) Line 205:206 The correlated binomial model should be expressed as either
N_i ~ poisson(lambda); C_ij ~ binomial(N_i,p_ij,rho); p_ij ~ beta(alpha,beta)
or
N_i ~ poisson(lambda); C_ij ~ beta-binomial(N_i,p_ab, rho)
where the correspondence between the two formulations is
p_ab = alpha/(alpha + beta) and rho = 1/(alpha+beta + 1)
I do not believe it is correct as you have written it. I’d suggest the first variation with the derived parameters expressed in the text.
3) Line304: I found this line hard to make sense of. The text reads very well overall, but I found this whole paragraph difficult to read. In terms of contrasting and assessing alternative estimation methods, it is probably one of the most important paragraphs in your paper, so you might like to reassess its structure.
4) None of your models included any spatial covariates (i.e., you assumed homogeneous habitat suitability). Model comparison is used in the SECR model to choose among candidate distributions (i.e., to test the inclusion of overdispersion parameters), but it could also be used to assess the merits of including any available predictors. Did you have any suitable covariates? This should be discussed.
5) Failing to account for imperfect detection is identified as a problem in the introduction, yet you assume perfect detection throughout your models (e.g., Line 230). Even though the weaker models appear to overestimate the density (suggesting lack of independence rather than thinned observations), the rationale your modelling choices concerning imperfect detection should be discussed/justified.
6) Line 321-322 Do you have any further thoughts as to why the beta-binomial model produces such large estimates? You mentioned the low detection rates, but is overestimation a known issue in this low-rate scenario? What was the estimated correlation, did it seem reasonable?
7) I found the plots difficult to study because the leopard and dhole results are spread across different plots. It would be helpful to see the plots grouped by species to make a direct comparison of the estimates for the different models. In particular, as side-by-side comparison of the closeness of the SECR and STE estimates for leopards, and the divergence of STE and Beta-Binomial models for Dholes would emphasise your main results. I don’t think that including both abundance and density adds anything useful to the plot since one is a re-scaling of the other.
8) The inclusion of MLE and Bayesian SECR estimates seems unnecessary. Unless it is your goal to draw attention to the similarity of the results, I would suggest it is sufficient to state that estimates are similar, presenting just one and reallocating the figure space.
9) You have presented results for various percentage-adjusted viewshed estimates, but these results are not discussed. I think they should be.

·

Basic reporting

The article is well written and balanced with the correct use of language.

Experimental design

The study is designed well in that the camera traps were randomly placed and the estimates of dhole population (unmarked) from various methods were compared with that of the leopard (marked) for checking their reliability, using appropriate analytical tools. However, I am concerned with the duration of camera trapping and the method of selecting trap stations. I am not sure whether 16 to 18 days of camera operation is adequate to capture all dholes in the area. The authors need to provide an empirical basis as to why this duration was chosen. I am tempted to ask how camera-trap stations were selected within a chosen grid (4 sq. km) – e.g., based on sign density, proximity to a den, etc. It would also be helpful for future study replication if there is an explanation on why a particular grid size of 4 sq. km was chosen and only 34 out of 80 grid cells were selected for camera stationing.

Validity of the findings

The findings are appropriate, but there is a need for stating the population figures estimated by site-based models since it was hard to understand population figures from figure 3. Please discuss why REM or REST models could not be applied to camera-trap data for dholes in this study. What precluded their use?

Additional comments

Here are some specific comments:

Line 72: Considering removing the comma after 2018b.
Line 156: fit or fitted?
Line 252: The supplementary file 3 was not available for viewing; the authors may have seemingly forgotten to upload it.

---

## Round 0.2 · accepted · Accept

I thank the authors for the careful consideration of the referee's critiques and general comments, and for clearly explaining the changes made to address these (in the response letter and tracked-changes MS). I am satisfied with the revisions, and congratulate the authors on a fine contribution.

Also, as a camera-trapper myself, I do understand the limitations imposed by real-world logistics, including the ever-challenging problem of camera thefts!

We appreciate you submitting your work to PeerJ, and hope you consider the journal again for future work in this area.